# The Fourth Survey on the Activity of Human Milk Banks in Italy

**DOI:** 10.3390/nu17162663

**Published:** 2025-08-18

**Authors:** Giuseppe De Nisi, Guido E. Moro, Sertac Arslanoglu, Amalia M. Ambruzzi, Enrico Bertino, Augusto Biasini, Claudio Profeti, Guglielmo Salvatori, Paola Tonetto, Pasqua Anna Quitadamo, Nicoletta Danese

**Affiliations:** 1Board of Directors of AIBLUD (Italian Association of Donor Human Milk Banks), 20126 Milan, Italy; giudenisi@gmail.com (G.D.N.); amaliaambruzzi@virgilio.it (A.M.A.); augustoclimb@gmail.com (A.B.); claudioprofeti@gmail.com (C.P.); 2Division of Neonatology, Department of Pediatrics, School of Medicine, Istanbul Medeniyet University, Istanbul 34720, Türkiye; sertacarslanoglu@gmail.com; 3Neonatal Care Unit, University of Turin, City of Health and Science of Turin, 10126 Turin, Italy; enrico.bertino@unito.it (E.B.); paola.tonetto@unito.it (P.T.); 4Neonatal Care Unit, Bambino Gesù Children’s Hospital, IRCCS, 00168 Rome, Italy; salvatori.guglielmo@tiscali.it; 5Neonatal Care Unit, IRCCS “Casa Sollievo della Sofferenza”, 71013 San Giovanni Rotondo, Italy; pasquaq@tiscali.it; 6Neonatal Care Unit, AULSS 8 Berica, 36100 Vicenza, Italy; nicoletta.danese@aulss8.veneto.it

**Keywords:** human milk, donor milk, human milk banks, preterm infants feeding, breast milk, infant nutrition

## Abstract

Background: Scientific literature confirms the benefits of mother’s own milk (MOM) for both term and preterm infants. The feeding of pathological newborns, in particular the very low birth weight infants (VLBWIs), is dependent on human milk. When MOM is not available, pasteurized donor human milk obtained from a recognized Human Milk Bank (HMB) is the best alternative. Research aims: This survey aims to evaluate the activity of human milk banks (HMBs) in Italy in the years 2023–2024. Methods: Following the previous three surveys performed in 2012, 2016, and 2022, a fourth survey related to 2023 and 2024 was planned in the year 2025. A questionnaire was sent to the 44 HMBs officially operating in Italy, with questions regarding their management and activity, in order to collect national-level data. Results: All 44 Italian HMBs (100%) responded to this survey. The collected data confirm the results of the previous surveys, confirming an optimal adherence to the Italian Ministerial Guidelines. Almost all the HMBs (96%) apply the principles of self-control and the HACCP system, while the home milk collection service still requires improvement. Only 68% of HMBs organize collection and transport of the donated milk from the donor’s home to the bank. In addition, this survey shows the spreading of computerization in the management of the activities of HMBs: 36.4% make use of specific software that could lead to a greater availability of donor human milk for the neonatal centers in the future. The number of donors and the amount of donated milk increased consistently compared to the previous years. Conclusions: In general, this survey shows an improvement in the results obtained in the three previous surveys, with a positive dissemination of the culture of human milk donation in Italy. The impressive response rate to the survey demonstrates the importance of a regular check-up of the activity of HMBs.

## 1. Background

### 1.1. Benefits of Human Milk

All infants should be fed their mother’s own milk. When MOM is lacking or insufficient, DHM obtained from a certified HMB is the best alternative. The benefits of human milk are known not only for the full-term infant, but also and especially for the preterm infants or those with specific pathologies [1,2].

Therefore, fresh MOM is essential for VLBWI, particularly in the early and most critical stages of life: it significantly improves the prognosis of these children by promoting physiological development and protecting their immature organism from many short- and long-term pathological conditions. When MOM is not available or is insufficient, the possibility of using donor human milk (DHM) appears strategic. The WHO [3,4], the AAP [5,6], and the ESPGHAN [7] recommend the use of banked human milk as the best alternative to mother’s milk, especially for VLBWI, in combination with maternal lactation support.

Donor human milk characteristics do not overlap with those of breast milk due to the processing treatment performed in HMBs. Despite this limitation it contains, unlike formula, species-specific substances valuable for physiological functions and for the protective effects on the immature organism of a preterm infant [8].

In this context, the particular nutritional and immunological composition of human milk makes it a unique and inimitable nutrient both for term and preterm infants [9]. Compelling evidence confirms the substantial benefits of human milk in the neonatal period, as prevention of necrotizing enterocolitis, reduced incidence of infections, better feeding tolerance, reduction in hospital stay, and cost savings [9,10,11,12,13,14].

### 1.2. The Situation of Human Milk Banks (HMBs)

HMB is a service aimed at selecting, collecting, checking, treating, storing, and distributing human milk, voluntarily donated and used for specific medical indications (primarily to satisfy the needs of VLBWI). The activity of HMBs and milk donation must be non-profit. The European Directorate for the Quality of Medicine & Healthcare (EDQM) recently established that donor human should be included in the group of substances of human origin (SOHO), and according to SOHO regulation, no financial incentives to donate, but only compensation for donor expenses, are permitted. The mission of HMBs is based on the promotion and support of breastfeeding, on the dissemination of the culture of donation, and the appropriate use of donor milk. In this way, HMBs contribute to the promotion of health and the reduction in infant morbidity and mortality.

### 1.3. The Italian Context

The first survey was conducted in Italy in the year 2000 with the purpose of evaluating the number of active HMBs in our country, their distribution, and their organization. It highlighted a very low number of banks (18), an unbalanced distribution, and a high variability in their organization. In 2002, a specific Working Group of the Italian Society of Neonatology published the first “Italian Guidelines for the Establishment and Operation of a Donor Human Milk Bank”, followed by updated editions in 2007, 2010, 2012, and by specific recommendations in 2021 [15].

The Italian Association of Donor Human Milk Banks (“Associazione Italiana Banche del Latte Umano Donato, AIBLUD”) was founded in 2005. All the 44 human milk banks (HMBs) active in Italy today (Figure 1) are members of AIBLUD, and their management is consistent with the Italian Guidelines. Technical updating and scientific activity are the priority objectives of the association.

The Italian Guidelines [15], and the results of the 2012, 2016, and 2022 [16] surveys on Italian HMBs, led to the opening of new banks with an improvement of their organization, and the closure of inefficient ones. In 2013, the Italian Ministry of Health published the National Recommendations on HMB in the Official Gazette of the Italian Republic [17].

## 2. Research Aim

The current survey, planned by AIBLUD, aims to evaluate the activity of HMBs operating in Italy in the period 2023–2024.

## 3. Methods

### 3.1. Research Design

All the banks operating in Italy in the year 2025 were invited to participate in this survey. They have been highly responsive in previous surveys and have shown great interest in presenting the most recent data on their activities.

No ethical approval was required for the carrying out of this survey.

### 3.2. Sample

The 44 banks active in Italy in 2025 were involved, and all HMBs participated in the survey. Therefore, our study refers to 44 HMBs (Figure 2). In the previous surveys, the percentage of participation was lower: 87.5% in 2012, 91.4% in 2016, and 90.2% in 2022.

### 3.3. Data Collection

In the year 2025, a detailed questionnaire (Table 1) was used to obtain data related to the activities of Italian HMBs. The data collection was performed with a Google form sent by e-mail to the persons responsible for the banks. The questionnaire consisted of 39 items: equipment, management, regionalization, number of donors, volume of human milk collected, and other information referring adoption of the HACCP system [18]; milk transport from the donor’s home to the bank; and regional activity as a hub center related to the years 2023 and 2024.

### 3.4. Data Analysis

The number of donors, the volumes collected in liters of donor milk, and the percentage of responses to the questions posed in this survey are compared with those of the survey performed in 2016. The results of the present survey were not compared to the results of the 2022 survey, because they were heavily influenced by the COVID-19 pandemic.

## 4. Results

The results obtained from this survey are presented in Table 2, Table 3, Table 4, Table 5 and Table 6.

Table 2 shows a satisfactory equipment supply (pasteurizers and freezers) and the frequent use of milk pools: 66% from single donors and 41% from multiple donors. The question on data collection method highlighted that only 36% of the banks use dedicated software, while 64% of HMBs rely on paper forms or spreadsheets. The application of the principles of the HACCP system (96%) has improved from previous surveys. On the other hand, the collection of donor milk at home with a targeted service organized by the banks (64%) still requires improvement, even if almost all the banks provide the material for donation to the breastfeeding mothers (mainly extraction pumps and bottles).

Table 3 shows the attention paid to infectious tests of prospective donors with a group of 10 banks using the NAT (Nucleic Acid Test) combined with serology. The data on bacteriological monitoring of donor human milk, both before and after pasteurization, remains high as in previous surveys. The answer to the question on Bacillus Cereus is interesting: 66% of the banks search for it, and all of them discard the milk if the test is positive.

Table 4 shows the data related to number of donors and volume of milk collected for the years 2023 and 2024, compared with the 2016 survey. The comparison with the third survey of 2022 [16] did not seem appropriate as it refers to the period of the COVID-19 epidemic. The volumes of milk collected and the average duration of the donation are similar in the two surveys.

The number and type of infants fed with donor human milk were also considered (Table 5). The data show that the main utilization of donor milk was for feeding very low birth weight (VLBW) infants and pathological infants over 2499 g. In recent years, Italy has seen a sharp decline in the number of live births, particularly in 2023 and 2024. The percentage of infants fed with banked human milk (Table 6) has increased significantly: 1.24% versus 0.90–1.05% in the previous surveys. The data referring to the number of VLBW infants receiving DHM remained quite constant during the years, with a percentage of 34.2% of all the VLBW infants born in Italy in the year 2024.

## 5. Discussion

National guidelines aim to standardize the activity of HMBs; however, verification of their implementation remains crucial.

A web-based questionnaire was developed by the European Milk Bank Association (EMBA) Survey Group [19] for distribution to the European HMBs. A total of 123 replies (response rate 57%) from 22 out of the 26 European countries were received.

The Brazilian Network of Human Milk Banks (rBLH Brazil) [20], established by the Brazilian Ministry of Health and the Oswaldo Cruz Foundation (FIOCRUZ) in 1998 [21], continues to monitor the activity of HMBs distributed throughout the Brazilian states: in 2025 there were 237 HMBs. In 2019, the 222 HMBs of rBLH Brazil collected 222,696 L of human milk.

A survey conducted in 2022 in Japan [22] involved 47 NICUs that had access to HMBs: 37 of 47 (78.9%) NICUs answered the questionnaire. The most common indications for DHM were gestational age less than 28 weeks (78.3%) and birthweight less than 1500 g (100%).

A survey conducted in the United States revealed the lack of standardization in the activity of HMBs belonging to the Human Milk Banking Association of North America (HMBANA) [23].

In Italy, AIBLUD in 2012 performed the first national survey, which involved all the Italian HMBs in order to monitor their activities. In 2013, AIBLUD, together with the Italian Ministry of Health, published the “Italian National Recommendations for the Organization and Management of HMBs as a Tool for the Protection, Promotion, and Support of Breastfeeding” in the Official Gazette of the Italian Republic [17]. Further combined surveys (AIBLUD and Italian Ministry of Health) were conducted in the years 2016 and 2022 [16].

An interesting result coming out from the present survey was the discovery that 14 banks (32%) provided DHM to other hospitals. This “hub” function leads to a reduction in costs and a greater diffusion of the culture of donation and the use of human milk for the newborns at risk.

Since 2018, AIBLUD has been organizing regular training courses for the management of HMBs (2–3 courses per year in different regions). These courses confirm that the activity of Italian milk banks is consistent with the indications of the Italian Guidelines.

In the past, the main recurrent problems were linked to the application of the principles of the HACCP system and to the home collection service of donor milk. Therefore, this survey evaluated these two critical points together with other data of the banks’ activity (number of donors, volumes of donor milk, number and type of infants fed with donor milk).

The number of Italian HMBs has increased compared to 2022 (seven new banks), particularly in Lombardy, Veneto, and Sicily, the regions with the highest number of banks. Southern Italy still has a limited number of HMBs operating in the territory. The lack of HMBs in the South of Italy represents an old gap due to many factors, like tradition, a culture less interested in human milk, lower economic resources, etc. But things are starting to change, as demonstrated by Sicily (a region with only one HMB till a few years ago), where four new HMBs opened in the last 2 years. Today, Sicily, with five HMBs operating locally, is the fourth region in Italy for the number of HMBs, after Lombardy with seven banks, and Veneto and Tuscany with six banks.

A fundamental task of an HMB is the collection and transport of human milk from the donor’s home to the bank. In the two-year period 2023–2024, in Italy, 36% of the banks did not guarantee this service.

Our guidelines strongly recommend the adoption of the HACCP system [18]. According to the results of the survey, 43 out of 44 of our banks (96%) use this system. This figure is significantly better than that recorded in the previous surveys.

Table 4 shows an increase of 12% in the number of donors from 2016 to 2024 (1336 in 2016 vs. 1520 in 2024), while the volumes of donor milk and the duration of the donation appear similar.

The number of newborns fed with donor human milk confirms the strong commitment of the banks’ staff. In 2023 and 2024, in Italy, there was a sharp decline in the number of births with a parallel decline in the total number of infants receiving DHM (4987 in 2016, 4695 in 2023, and 4735 in 2024) (Table 5). Despite this decrease, there was an increase in the percentage of infants fed with DHM that reached 1.28% of total births in 2024, higher than that of the previous years (Table 6). If we look only at the category of infants <1500 g, we can see that the percentage of VLBW fed with DHM has significantly increased in the period 2016–2023 from 29.3% to 36.2%, and has decreased to 34.2% in 2024 (Table 6). Italy is a country that has lost 2 million inhabitants in the last ten years, with a significant decrease in the number of births every year. These figures can explain the lower total number of infants receiving human milk from HMBs and the decrease in VLBW infants receiving DHM in the years 2023 and 2024

This figure shows that there is still a significant gap in the extensive use of donor milk in VLBW infants and in sick infants. Substantial scientific evidence supports the benefits of using human milk for the health of VLBW infants [9,10,11,12,13,14].

The implementation of the Italian Guidelines, together with the training courses organized by AIBLUD, has helped to open new HMBs in Italy: from 2018 to 2024, twelve new HMBs started their activity in Italy. The results of this survey show that it is necessary to periodically monitor the activity of HMBs, and AIBLUD is particularly involved in this surveillance.

The main strength of this study is the fantastic rate of participation by Italian HMBs: all 44 Italian HMBs (100%) responded to the survey.

A limitation of this study is its national coverage, which gives information on HMBs in Italy without the possibility to compare our national data with data from other countries in Europe. The next step should be an international survey in order to evaluate HMBs’ activity in the European countries with the highest number of HMBs, like Italy, France, Germany, and Spain.

## 6. Conclusions

This survey confirms the quality of HMB’s service and its usefulness for Italian health policy; it also highlights that the act of donation is part of the life of a community and must, therefore, face all social, economic, and health aspects.

However, in Italy, infants fed with donor milk are still too few: only 1.28% of live births. A lot of work still needs to be carried out to spread the culture of donation and use of human milk in the most vulnerable infants.

## Figures and Tables

**Figure 1 nutrients-17-02663-f001:**
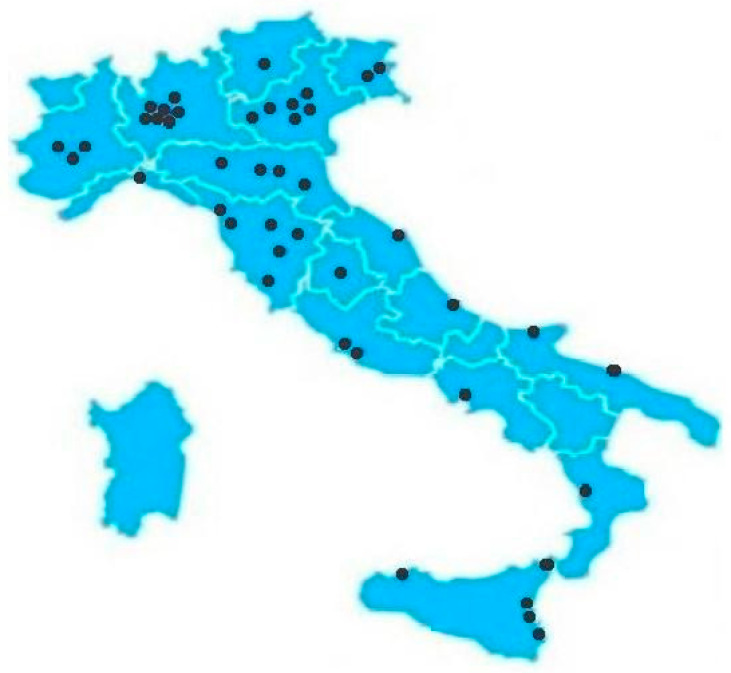
The distribution of the 44 human milk banks in Italy (2025).

**Figure 2 nutrients-17-02663-f002:**
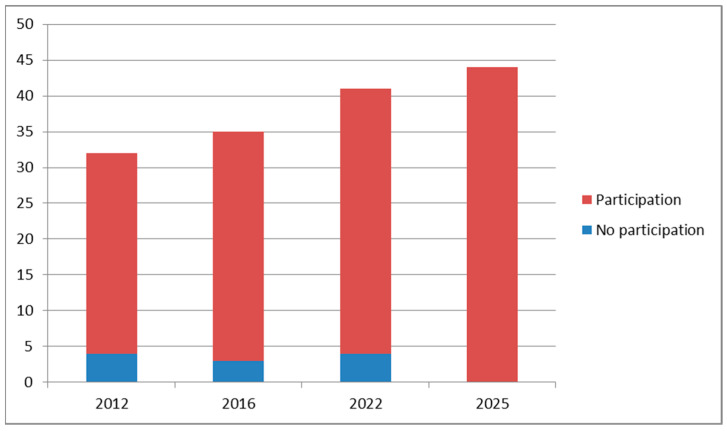
Comparison of HMB participation in the 4 surveys: 2012, 28 out of 32; 2016, 32 out of 35; 2022, 37 out of 41; and 2025, 44 out of 44.

**Table 1 nutrients-17-02663-t001:** The questionnaire sent in the year 2025 to the 44 human milk banks operating in Italy.

**NAME OF PERSON COMPLETING THIS QUESTIONNAIRE** **MILK BANK NAME and ADDRESS** **NAME OF RESPONSIBLE PERSONS (Manager and Clinical)**
**STATISTICS** Number of pasteurizers and freezers Use of pool of donor milk Donor’s screening and microbiological checks on donor milk Number of donors in years 2023 and 2024 Volume of donor milk collected in years 2023 and 2024 Average length of donation in years 2023 and 2024 Number of infants fed with donor milk in years 2023 and 2024
**HOME COLLECTION SERVICE** Transport service for home collection of donor milk and provision of material
**HACCP (HAZARD ANALYSIS AND CRITICAL CONTROL POINT)** Application of the principles of the HACCP system
**REGIONALIZATION OF THE SERVICE** Human milk bank with regional activity as a “hub” center

**Table 2 nutrients-17-02663-t002:** General management of Italian human milk banks.

General Management	No. Banks	%
Pasteurizers No. > 1	16	36
Utilization of Holder pasteurization	44	100
Freezers No. > 2	18	41
Temperature of storage ≤ 20 °C	44	100
Use of pools of milk from single donors	29	66
Use of pools of milk from multiple donors	18	41
System data collection with paper	22	50
System data collection with spreadsheet	6	14
System data collection with specific software	16	36
Milk supplied to other hospitals	14	32
Tracking from donor to recipient	44	100
Home collection service	28	64
Provision of material for milk collection at home	43	98
Use of HACCP * system	42	96

* HACCP = Hazard Analysis and Critical Control Point.

**Table 3 nutrients-17-02663-t003:** Donor’s screening and testing with microbiological checks on donor milk.

Screening and Microbiological Tests	N. Banks	%
Infectious tests on prospective donor performed at the beginning of donation	40	91
Infectious tests on the prospective donor accepted if they have been performed no later than three months before the beginning of donation	41	93
Type of infectious tests performed on the donor: serology only	34	77
Type of infectious tests performed on the donor: serology + NAT *	10	23
Bacteriological analysis on donor milk: only at the beginning of donation on unpasteurized milk	37	84
Bacteriological analysis on donor milk: at beginning of donation on unpasteurized milk, and random after pasteurization	13	30
Bacteriological analysis on donor milk: only random after pasteurization	30	68
Bacillus Cereus search on donor milk	29	66
Discarding donor milk if it tests positive for Bacillus Cereus	44	100

* NAT = Nucleic Acid Test.

**Table 4 nutrients-17-02663-t004:** Number of donors and volumes of donor human milk collected in the present survey compared to the survey performed in 2016. We decided not to show the results of the survey performed in 2022 because they were deeply influenced by COVID-19 pandemic.

Year	2016	2023	2024
Number of donors	1336	1431	1520
Volume of donor milk collected (L)	9181	9120	9631
Average volume of milk collected per donor (L)	6.9	6.4	6.3
Average length of donation (days)	142	114	117

**Table 5 nutrients-17-02663-t005:** Number of infants fed with donor human milk in Italy according to birth weight.

Birth Weight	2016	2023	2024
<1500 g	1299	1239	1139
1500–2499 g	1447	1729	1813
>2499 g (in NICU *)	1049	1034	1198
Healthy term infants **	1192	693	585
Total	4987	4695	4735

* NICU = Neonatal Intensive Care Unit. ** healthy term infants: term infants needing an increased intake of liquids in the first 48–72 h of life due to hyperbilirubinemia, hypoglycemia, or excessive weight loss.

**Table 6 nutrients-17-02663-t006:** Number of live births and VLBW * infants fed with donor human milk in Italy.

Year	Total Number of Live Births in Italy	N. of Infants Fed with Donor Milk	Total Number of VLBW * Infants Born in Italy	N. of VLBW * Infants Fed with Donor Milk
2016	474,925	4987 (1.05%)	4749	1299 (29.3%)
2018	442,676	3984 (0.90%)	4427	1014 (26.7%)
2019	421,913	4180 (0.99%)	3797	993 (27.3%)
2020	404,260	3936 (0.97%)	3638	946 (19.9%)
2023	379,890	4695 (1.24%)	3419	1239 (36.2%)
2024	370,000	4735 (1.28%)	3330	1139 (34.2%)

Source: Italian Ministry of Health (https://www.salute.gov.it, accessed on 25 June 2025). * VLBW: very low birth weight.

## Data Availability

The original contributions presented in this study are included in the article. Further inquiries can be directed to the corresponding author.

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
