# Peer review of "The Fourth Survey on the Activity of Human Milk Banks in Italy"

_nutrients, 2025, doi:10.3390/nu17162663_

Round 1

Reviewer 1 Report

Comments and Suggestions for Authors

I am grateful for the trust placed in me to evaluate this research, which is part of my healthcare practice. 
It is a study in which the importance of the subject is well described in the introduction, but it is true that some possible considerations to be taken into account are indicated:
- improve the transition between paragraphs (eg: In this context ....; Therefore .....), this would consolidate a better understanding for the scientific world, in this sense comment that begins with the description of the general benefits of breast milk and then begins a paragraph of donors and international guidelines, it seems important to add a link between these issues. It is advisable to start with 1) benefits of BF, followed by 2) the situation of milk banks and 3) addressing the Italian context. It is advisable in the introduction to value broad paragraphs, as these can be divided into key ideas improving the vision of the manuscript, as an example it is indicated to refer to a definition of human milk bank, from the indication of its functions. A significant number of bibliographical references are included in this section, giving solidity and strength to the introduction.

Within the methodology, it is well explained and appropriate to the type of study (questionnaire), although it is true that the section on Ethics is missing, as the number granted by the committee should be provided.

In the Results section, we find a number of tables that clarify the study itself and where the content of the research is made explicit.

Regarding the discussion, it seems important the international comparison that is presented, as well as the update of data at national level and how the weakness of coverage in Southern Italy is expressed.
However, it is important to clarify that this section should be reviewed in detail, as it is part of the results of the study, which should appear in its own section.
It is also important to comment that some issues that are indicated could have a more in-depth analysis, for example: The percentage of VLBW fed with DHM increased to 36.2% in 2023 and decreased to 34.2% in 2024... In this sense, what could the decrease be due to, and is it related to logistical availability, or a change in the birth rate? Be careful as there is an (undue) repetition of issues already discussed in the introduction, and reiterated in this paragraph .... The use of DHM leads to a reduction in costs and a greater diffusion of the culture of donation"...already discussed previously.
Some modifications to the English are recommended:
- ‘A huge scientific evidence...’ → Better: ‘Substantial scientific evidence supports...’
- ‘we are still a long way from...’ → Better: ‘there is still a significant gap in...’ → Better: ‘there is still a significant gap in...’.
At the end of the paragraph, it would be useful to take into account limitations of the study, such as (national) coverage...among others....

Overall, it is a relevant study that may be of interest to an international audience, and I hope that you will find the indications I provide for an improvement of the manuscript appropriate.

Reviewer 2 Report

Comments and Suggestions for Authors

This is a well written paper about an adequately designed questionnaire based survey on the performance of the Italian Human Milk Banks. It also presents the trends in performance over a period of 8 years with several recommendations for further improvement.

The readability of the paper could be improved by redrawing Fig. 2 eg. using barchart instead of piechart and illustrating the increase in number of HMBs. I would also recommend redrawing all the (central alligned) tables.  

Reviewer 3 Report

Comments and Suggestions for Authors

Thank you for the opportunity to review this interesting article.

Impressive response rate - well done.

Possible put a recommendation in conclusion in abstract.

Line 57/8 sentence is nt clear and may be best to be two sentences.

Line 62 is this for terms or preterm. Should this be part of previous paragraph. Make links here and explain more.

Line 66 is repetitive as said this earlier. Integrate where written previously.

Flow could be more logical as currently disjointed.

Why must this be non profit needs explaining more.

Line 76 - need to state what the survey was about. First survey for what?

This and the next paragraph need linking together. A paragraph should be more than two sentences.

Line 82 needs developing and integrating. Same comment for next paragraph - link work more.

Line 82 may fit better with line 91.

Line 104 typo.

Reviewer 4 Report

Comments and Suggestions for Authors

Here is the review

Round 2

Reviewer 4 Report

Comments and Suggestions for Authors

The authors significantly improved the manuscript. It should be published in the present form